# Cebranopadol as a Novel Promising Agent for the Treatment of Pain

**DOI:** 10.3390/molecules27133987

**Published:** 2022-06-21

**Authors:** Wojciech Ziemichod, Jolanta Kotlinska, Ewa Gibula-Tarlowska, Natalia Karkoszka, Ewa Kedzierska

**Affiliations:** Chair and Department of Pharmacology and Pharmacodynamics, Medical University of Lublin, 4a Doktora Witolda Chodźki Street, 20-400 Lublin, Poland; jolanta.kotlinska@umlub.pl (J.K.); ewa.gibula-tarlowska@umlub.pl (E.G.-T.); natalia.e.karkoszka@gmail.com (N.K.); ewa.kedzierska@umlub.pl (E.K.)

**Keywords:** cebranopadol, pain, opioids, nociceptin, acute-pain, chronic-pain, orphanin

## Abstract

Opioids are used to treat pain, but despite their effectiveness, they possess several side effects such as respiratory depression, tolerance and physical dependence. Cebranopadol has been evaluated as a solution to this problem. The compound acts on the mu opioid receptor and the nociceptin/orphanin receptor and these receptors co-activation can reduce opioid side-effects without compromising analgesia. In the present review, we have compiled information on the effects of cebranopadol, its pharmacokinetics, and clinical trials involving cebranopadol, to further explore its promise in pain management.

## 1. Introduction

Opioid drugs, which act on the mu opioid (MOP) receptors, are among the most powerful and effective available analgesics. The usefulness of these drugs in the treatment of acute post-traumatic, postoperative, or chronic pain is well known. Therefore, in the last few decades, the number of patients with non-cancer pain that have been prescribed strong opioids has increased. However, treatment with opioids also comes with many side effects, including drowsiness, confusion, nausea, constipation, breathing depression, euphoria and finally abuse. Consequently, in employing MOP agonists in the treatment of pain, addiction and opioid tolerance pose a significant challenge. Due to its euphoric effect, one of the most commonly administered MOP agonists—morphine—is also a substance used for non-medical purposes. Apart from addiction, an additional problem in the case of the employment of morphine in medicine is the phenomenon of tolerance, which is associated with the need to increase the amount of the taken substance in order to achieve the original effect [1]. Particularly significant is the fact that, in humans, this process is very dynamic and grows heterogeneously with a rapid development in the analgesic and euphoric effects, slower for depressive effects on the respiratory center, and little or no effects such as inhibition of gastrointestinal peristalsis or myosis [2,3,4]. However, presently it is known that delta opioid (DOP) agonists may also evoke analgesic effects. Many authors indicate that selective activation of DOP has great potential for the treatment of chronic pain [5,6] with ancillary anxiolytic- and antidepressant-like effects [7]. This is particularly relevant in view of the frequent association of anxiety and mood disorders with chronic pain [8]. Moreover, DOP agonists typically show reduced adverse effects, notably regarding abuse potential and respiratory depression. Likewise, selective kappa opioid (KOP) agonists produce a strong analgesic effect without causing addiction. A major disadvantage in their widespread usage as analgesics, however, is their dysphoric potential [9].

Due to the limited activity of classic opioids in the treatment of neuropathic pain, their strong addictive potential and their wide range of side effects, new compounds with opioid-like effects have been the subject of research [10]. A significant achievement in this process was the identification of the nociception opioid peptide receptor (NOP), which in humans is encoded by the opioid receptor-like-1 (ORL-1) gene. This is a G protein-coupled receptor with a high homology to opioid receptors, but without the ability to bind opioid ligands [11]. One year after the discovery of this receptor, the endogenous NOP ligand was identified and named “nociceptin/orphanin FQ” (N/OFQ). This peptide was found in the central (CNS), as well as in the peripheral nervous system (PNS), where it presumably modulates nociception [12,13]. However, in contrast to morphine and other opioids that are used to alleviate pain, the role of this peptide in nociception is not straightforward. Previous studies with rodents indicated that N/OFQ administrated intrathecally (i.t.) enhances morphine—or electroacupuncture-induced analgesia [14,15,16], and it produces antinociceptive synergy with morphine in the model of neuropathic pain [17]. Moreover, it has been observed that the systemic coactivation of MOP and NOP receptors produced a synergistic antinociception in primates [18,19] without the side effects typical for classic opioids [20,21]. In rodents, NOP receptor activation has been shown to counteract the MOR agonist-mediated development of tolerance, addiction and physical dependence [21]. Furthermore, the administration of exogenous N/OFQ, concomitantly with morphine, attenuates the development of morphine tolerance, without impact on the basal and morphine nociceptive responses [11]. These studies strongly support the therapeutic potential of mixed MOP/NOP agonists as innovative analgesics.

Therefore, recent research has focused on low-selectivity and multifunctional “mixed ligands” in attempts to generate new analgesics. The consequence of this research is the introduction of cebranopadol, a first-in-class potent analgesic agent with agonistic activity that targets NOP and opioid receptors [22].

## 2. Cebranopadol as an NOP and Opioid Receptors Agonist

Cebranopadol is a new and promising agent in the treatment of pain. It is a spiroindole derivative of the benzenoid class with the UPAC approved name: 6-fluoro-N,N-dimethyl-1′-phenylspiro[4,9-dihydro-3H-pyrano[3,4-b]indole-1,4′-cyclohexane]-1′-amine (Figure 1). It should be emphasized that this compound possesses a unique mechanism of action as a mixed NOP/opioid receptors agonist [23] of single molecule size characterized by high permeability into the CNS [24]. Previously, it has been established that cebranopadol acts as a full agonist of the MOP and DOP, as well as a partial agonist of the KOP and NOP [10,24].

## 3. Pharmacokinetics and Biological Availability of Cebranopadol

During the research and the introduction of a new drug to treatment, it is an important issue to evaluate the duration of its action. After intravenous (i.v.) administration of cebranopadol at the dose of 12 mg/kg, the time of action was 7 h, although the maximum possible effect (MPE) was 10%. In comparison, the activity of morphine and fentanyl was decreased after 180 min and 30 min, respectively. Following the administration of 55 mg/kg of cebranopadol, the activity lasted for at least 9 h and the maximum possible effect (MPE) was 52% [10]. In addition, cebranopadol was rapidly absorbed and extensively distributed, while the oral availability was estimated as 13–23%. Other pharmacokinetic parameters are summarized in Table 1 [10].

In a similar evaluation conducted by Rizzi et al., antinociceptive activity of cebranopadol and fentanyl occurs in the same dose ranges (0.01–1 mg/kg administered i.v.) but with different times of action. Fentanyl revealed a peak of analgesia at 5 min post-injection, and it lasted for 90 min at the highest doses. Cebranopadol, despite having slower onset (the maximum of the activity was established after 30 min), had activity that lasted for 120 min [23].

Kleideiter et al. [25] assessed the clinical pharmacokinetics characteristics of cebranopadol in six phase I clinical trials in patients with chronic low back pain versus healthy control. They observed that, after the administration of an immediate-release (IR) form of cebranopadol, maximal concentration of the drug in plasma [C max] occurred after 4–6 h. The study also revealed that the drug has a long time of half-value duration [HVD], which was determined at 14–15 h and a terminal phase half-life in a range of 62–96 h. Kleideiter et al. [25] established that, after multiple once-daily doses of cebranopadol to patients, an operational half-life of 24 h was found to be an essential factor for determining the pharmacokinetic properties after multiple administration. A steady state was accomplished after two weeks (approximately value), while AF (accumulation factor) was established as 2 and the peak–trough fluctuation (PTF) was estimated as 70–80%. These researchers also demonstrated a dose proportionality at a steady state for a broad range of doses (200–1600 μg) and found that the effect of these covariates is not clinically significant due to the broad therapeutic window of cebranopadol. The expected therapeutic doses have been estimated at 200–600 μg/day, which should be achieved after an uptitration period. Therefore, there is no necessity to adjust the dose for the individual as it is with morphine [25].

Interestingly, scientists from the research group of Łebkowska–Wieruszewska evaluated the pharmacokinetics of cebranopadol at the dose of 200 μg/kg in rabbits after subcutaneous (s.c.) administration. Subsequently, the scientists measured the concentration of cerbanopadol in blood samples which were withdrawn at 15, 30 and 45 min and 1, 1.5, 2, 4, 6, 8, 10 and 24 h after the administration. Accordingly, cebranopadol was quantifiable from 0.25 to 10 h, while mean Cmax and Tmax were established as 871 ng/mL and 0.25 h, respectively. The absorption was rapid, and the mean terminal half-life was 3.85 h [26] (Table 1).

Cebranopadol is currently formulated as an immediate-release product for oral usage in a clinical environment. After a single oral administration, it is characterized by a late tmax (4–6 h), with a long half-value duration (approximately 24 h). Furthermore, the similar parameters obtained after both i.v. and oral administration suggest a complete but slow absorption. These parameters are most likely related to the low solubility of the compound. Extended pharmacokinetic studies also suggest that cebranopadol can be used as a once-daily dose in the treatment of chronic pain. Moreover, in the case of elderly people and those who are also taking other medications, it seems to be particularly important that cebranopadol can be administered independent of diet and cytochrome P450 (CYP450) isozyme activity (1A2, 2A6, 2B6, 2C8, 2C9, 2C19, 2D6, 2E1 and 3A4/5). Published data confirm that these isoenzymes are not inhibited by cebranopadol at concentrations up to 250 nM [25].

## 4. Analgesic Effect of Cebranopadol

Cebranopadol induces analgesic, antiallodynic and antihyperalgesic properties in several rat models of an acute nociceptive, inflammatory, cancer and neuropathic pain [27,28]. In contrast to classical opioids, it has a higher analgesic potency in several models of neuropathic pain than in acute nociceptive pain [29]. In addition, in rodents, even at higher doses, cebranopadol has limited potential to produce opioid-type side effects, particularly physical dependence [30].

It seems that, for the analgesic effects of cebranopadol observed in rodents as well as in non-human primates, there is a synergistic effect between an NOP and an opioid receptor activation. This suggestion was supported in experiments with both NOP and opioid receptor antagonists [10]. Furthermore, the activation of NOP counteracts the respiratory depressant action associated with opioid receptor stimulation, since the NOP receptor antagonist J-113397 potentiated the respiratory depressant effects of cebranopadol in rats [10]. Similar mechanisms might be involved in analgesia observed in humans where cebranopadol displayed less respiratory depression than the MOP selective agonist fentanyl [23]—as has been suggested by Tzschentke et al. [31]. In fact, via concurrent NOP activation, its desirable effects (analgesia) are increased, while its unwanted actions (respiratory depression, physical dependence) are counteracted. Therefore, this drug may greatly contribute to decreased opioid abuse (Table 2).

## 5. Cebranopadol—Abuse Potential and Drug Dependence

To evaluate the cebranopadol dependence potential, Tzschentke et al. conducted a naloxone-induced withdrawal test. In the experiments, no jumps caused by withdrawal of the drug up to a single dose of 8 μg/kg were observed. Furthermore, cebranopadol revealed a low physical dependence potential [31]. According to the research presented above, the cessation of the administration of cebranopadol after four weeks induced a body weight loss which was not dose-dependent. A slight increase in the withdrawal score was also noticed, but only in the group treated with a small dose of this compound [31]. Similar results were obtained in rats both experiencing a spontaneous and naloxone-precipitated withdrawal. These results suggest the lower potential of cebranopadol to produce physical dependence than morphine, which is presumably related to the stimulation of NOP [31].

Subsequently, the aforementioned observations were widened by using both wild-type mice and mice lacking the NOP receptor gene (NOP −/−). During the experiments, cebranopadol or morphine were administered twice a day for five days in increasing doses (Table 3).

On the fifth day of the experiment, two hours after the morning injection, a withdrawal syndrome was accelerated by the administration of naloxone (at a dose of 10 mg/kg). It was established that the equi-effective to morphine analgesic doses of cebranopadol in the wild type of mice induced an opioid-like physical dependence similar to morphine. The effects of the treatment of both NOP (+/+) mice and NOP (−/−) mice with morphine caused similar effects (in mice NOP (+/+) and NOP (−/−), whereas a stronger dependence on cebranopadol was observed in NOP (−/−) mice than in NOP (+/+). This suggests that the activation of NOP receptors can decrease the possibility of cebranopadol to produce opioid-like physical dependence. The simultaneous activation of both NOP and opioid receptors can be an effective pharmacological strategy to generate potent analgesics with low ability to produce physical dependence [37]. Furthermore, such outcome may suggest that NOP agonists (including cebranopadol) can be used to treat substance use disorders.

There are reports indicating that NOP and opioid receptors agonists can be used particularly in treating opioid addiction. According to the in vivo research conducted by Guglielmo et al., cebranopadol significantly reduced both the operant response for cocaine and cocaine self-administration in a dose-dependent manner (25 μg/kg; 50 μg/kg before sessions). Moreover, during the tests, neither the influence of cebranopadol on locomotor activity nor inactive lever responses were observed. Cebranopadol also did not influence self-administration of natural and highly palatable food like sweetened condensed milk [38]. Similar conclusions were drawn by Shen et al. [39]. Subsequently, Wei et al. examined the influence of cebranopadol on cocaine kinetics. In these experiments, no significant effects of cebranopadol on cocaine kinetics were detected [40]. However, despite the fact that cebranopadol decreases cocaine self-intake, Wei et al. indicated that cebranopadol at the dose of 50 ug/kg potentiated cocaine-induced hyperactivity [40]. According to Shen et al., the mechanism of action of cebranopadol on cocaine self-administration is based on two possibly independent mechanisms involving NOP and the classical opioid receptor, as only the simultaneous blockade of both these pathways determines the inhibition of cocaine self-intake [39].

## 6. Safety and Side Effects of Cebranopadol Administration

Cebranopadol administration is considered safer therapy than typical opioid-drug administration. The safety of cebranopadol was examined during in vitro and in vivo experiments. In the comprehensive research conducted by Linz et al. [10], the safety of the compound was evaluated in a rotarod test (which measures the balance, coordination, physical condition, and motor-planning of the rodent). These observations indicated that cebranopadol administered i.v. at effective doses (4, 8, 16 mg/kg) did not influence motor coordination, in contrast to effective doses of morphine (2.7 and 8.9 mg/kg). Despite this fact, it was observed that higher doses of cebranopadol administrated orally can evoke hyperactivity. Indeed, Linz et al. indicated that the dose of 75 as well as 100 μg/kg induced significant locomotor hyperactivity [10].

Due to the respiratory depression evoked by most classical opioids, the influence of cebranopadol on the respiratory system in conscious freely moving rats was evaluated. In this experiment, after administration of 4, 8 or 16 μg/kg of cebranopadol, the respiratory rate and tidal volume were not affected. Moreover, the compound did not significantly influence minute volume, peak inspiratory, expiratory flows, inspiration and expiration times. Furthermore, it did not significantly change the calculated airway resistance index. These outcomes distinguish cebranopadol from typical opioid drugs [10].

Further evaluation of cebranopadol also confirmed the safety of this compound. A non-randomized, multiside, open-labeled and single-arm clinical trial was conducted on patients who suffered from moderate to severe cancer-related pain, who had finished a double-blind trial in which cebranopadol was compared to PR (prolonged release) morphine. Herein, it was reported that cebranopadol was well tolerated and considered safe during prolonged treatment of up to 26 weeks. The tested range dosages were 200–1000 mg per day. Moreover, it was reported that switching from morphine PR to cebranopadol was successful in terms of analgesia, as well as being safe and well tolerated [41].

Therefore, cebranopadol, when compared to morphine, is more potent, produces longer lasting antinociceptive effects, and displays a lower respiratory depression and lower tolerance liability. The above data indicate that cebranopadol may become an innovative analgesic.

## 7. Potential Use of Cebranopadol

As the use of opioids in treatment is gradually being phased out, cebranopadol may become an alternative to the usage of classic opioids. The fact that the compound has a non-selective mechanism of action and can act as an opioid agent and as an NOP agonist generates many opportunities for its usage [20,21]. Research confirms that cebranopadol can be used to counter various types of pain, including: inflammatory, cancer-related, chemotherapy-induced, visceral, tonic, chronic and the pain induced by thermal and chemical stimulations. Furthermore, cebranopadol can be administered in the treatment of polyneuropathic pain, which distinguishes it from the opioids [10]. This activity is attributed to the cebranopadol agonistic properties to NOP receptors. Moreover, there are high expectations of using cebranopadol in the treatment of addiction. As described above, cebranopadol, due to its NOP and MOP agonistic activity, decreased cocaine intake in a dose-dependent manner [39]. Moreover, there is no significant evidence that cebranopadol induces opioid-like physical dependence. This is due to its ability to stimulate NOP receptors [31]. We can therefore presume that cebranopadol could be used as a prototypic drug in a pharmacological strategy to generate potent analgesics with reduced liability to physical dependence.

## 8. Conclusions

Cebranopadol is a new promising compound with a mixed mechanism of action, since it acts not only as an opioid, but also as an NOP agonist. This non-selective activity gives a new opportunity for the treatment of chronic moderate to severe pain. Furthermore, it is characterized not only by its analgesic activity, but also by providing a protective action against side effects typical for opioids. Furthermore, cebranopadol does not influence motor coordination—compared to morphine. It also did not significantly influence respiratory parameters, and it does not change significantly calculated airway resistance indexes. These effects distinguish cebanopadol from the typical opioids; hence, it can have potentially a wide range of application. It can be used not only for the treatment of pain, but also to counter addiction. Therefore, it seems that cebranopadol, as a new generation analgesic drug, may become an alternative to the usage of classic opioids.

## Figures and Tables

**Figure 1 molecules-27-03987-f001:**
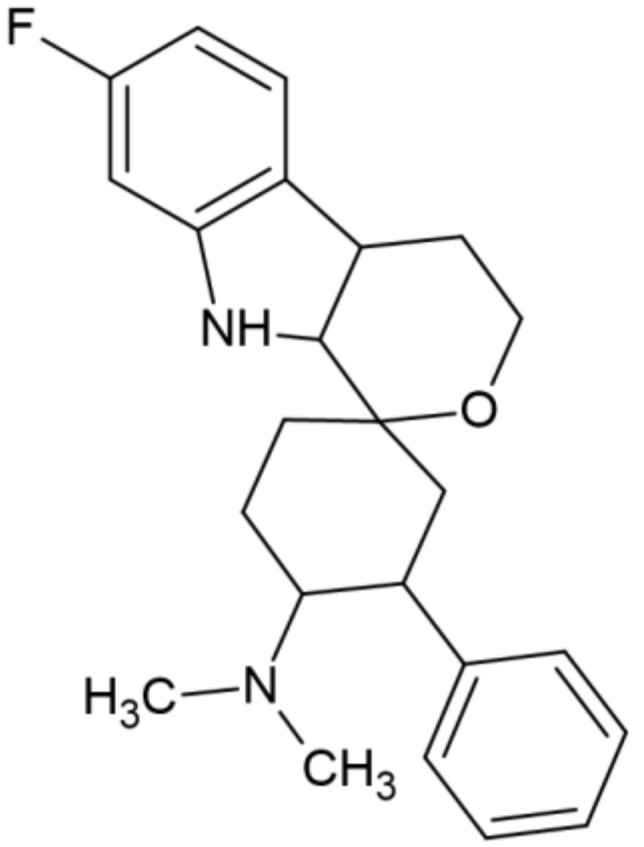
Structure of cebranopadol.

**Table 1 molecules-27-03987-t001:** Summary of selected pharmacokinetic parameters of cebranopadol in rodents. The table has been compiled and modified according to [10,27].

	Species/ Sex	*n*	Dose	Route of Administration	Parameter	Unit	Value [Mean ± S.D.]
Summary of pharmacokinetic parameters of cebranopadol after single i.v. administration to Sprague–Dawley rats [10].	Rats/ Male	*n* = 4	160 μg/kg	Intravenous	C0	ng/mL	22.8+/−1.01
					AUC	h × ng/mL	22.2+/−3.73
					T1/2	h	4.52+/−0.82
					CL	L/kg/h	7.37+/−1.38
					Vx	L/kg	47.1+/−5.34
							Value [Geom Mean (Min/Max)]
Summary of selected pharmacokinetic parameters after s.c administration of cebranopadol to rabbits [27].	Rabbits/ Female	*n* = 12	200 μg/kg	Subcutaneous	AUC (0-t)	Ug * h/L	1558 (1107/2365)
					AUC (0-inf)	Ug * h/L	1802 (1181/2737)
					T1/2	h	3.85 (2.50/7.07)
					Cmax	Ng/ml	871 (740/1150)
					Tmax^#^	h	0.25 (0.25/0.25)
					MRT	h	4.12 (2.69/7.1)

Abbreviations: C0, extrapolated concentration at the time of intravenous bolus administration (*t* = 0 h); AUC, area under the plasma concentration-time curve extrapolated to infinity; T1/2 terminal half-life; CL, total clearance; Vx, apparent volume of distribution during the terminal phase of disposition; AUC(0–inf), area under the curve from zero to infinity; AUC(0–t), area under the curve from zero to the last concentration; Cmax, maximum concentration; MRT, mean resident time; Tmax, time at which the Cmax occurs.

**Table 2 molecules-27-03987-t002:** The use of cebranopadol in alleviating different types of pain.

Animal Studies						
Pain Model	Species/ Sex	Route of Administration	Dose	Effect	Outcome	References
Induced visceral pain; colitis model	Mice/Male	Intravenous	4.6 μg /kg (half maximal effective dose, ED50)	Inhibition of spontaneous pain behaviours.	Cebranopadol displayed potent antiallodynic and antihyperalgesic effect in rodent models of visceral pain.	[32]
			2.2 μg /kg (ED50)	Inhibition of referred allodynia.		
			2.4 μg/kg (ED50)	Inhibition of referred hyperalgesia.		
Induced Visceral pain; pancreatitis model	Rats/Male	Intravenous	0.13 μg /kg (ED50)	Inhibition of abdominal tactile allodynia.		
Knee joint arthritis model	Rats/Male	Intravenous	0.8–8.0 μg /kg	Anti-hypersensitive effect.	Cebranopadol elicited potent, dose-dependent anti-hypersensitive effect in a rat model of arthritic pain (demonstrated in a weight-bearing test).	[33]
Pain induced by thermal and chemical stimulation.	Mice/Male	Intravenous	0.001–1 mg/kg	Antinociceptive effect.	Cebranopadol displayed highly potent antinociceptive effect in the tail withdrawal test. It was also very effective in inhibiting the nociceptive effect of formalin. Cebranopadol displayed higher analgesic potency against inflammatory rather than nociceptive pain.	[23]
Acute-tonic-chronic pain induced by thermal and chemical stimulation Neuropathic pain	Mice/Male	Subcutaneus (s.c)	10 mg/kg	Antinociceptive effect.	Cebranopadol displayed a significant antinociceptive activity in acute pain models, i.e., the hot plate, writhing, and capsaicin tests. It attenuated nocifensive responses in both phases of the formalin test and reduced cold allodynia in oxaliplatin-induced neuropathic pain model.	[34]
Bone cancer pain model	Rats/ Female	Intravenous	2.4, 8.0, 24.0 μg /kg	Increase in ipsilateral paw withdrawal thresholds.	Cebranopadol dose-dependently increased ipsilateral paw withdrawal thresholds.	[10]
Diabetic polyneuropathy model (streptozotocin, STZ induced)	Rats/Male	Intravenous	0.24, 0.8, 2.4 μg /kg	Inhibition of mechanical hyperalgesia.	Cebranopadol showed dose-dependent and significant inhibition of mechanical hyperalgesia at all doses tested.	
Mononeuroptic pain (Spinal Nerve Ligation, SNL model)	Rats/Male	Intravenous	0.24, 0.8, 2.4 and 8.0 μg /kg	Inhibition of mechanical hypersensitivity.	Cebranopadol showed dose-dependent inhibition of mechanical hypersensitivity.	
Temperature induced pain	Rats/ Female	Intravenous	17 mcg/kg (maximum effective dose)	Inhibition of heat nociception.	In the tail–flick test, cebranopadol induced dose-dependent inhibition of heat nociception.	
		Oral	80 μg /kg (maximum effective dose)			
**Human Research**						
**Patient Population**	**Age/** **Gender**	**Route of** **Administration**	**Dose Number of** **Subjects (N)** **Treatment Period**	**Primary Efficacy** **Endpoint**	**Outcome**	**References**
Patients with moderate-to-severe chronic-lower back pain (LBP) of nonmalignant origin	18 to 80 years of age Male/ Female	Oral	200 μg (*n =* 129), 400 μg (*n* = 127), 600 μg (*n* = 127) once daily treatment period: 14 weeks	The change from baseline pain to the weekly average 24 h pain during the entire 12 weeks of the maintenance phase and the change from baseline pain to the average 24 h pain during week 12 of the maintenance phase.	Cebranopadol demonstrated analgesic efficacy in patients suffering from moderate-to-severe chronic LBP, with and without a neuropathic pain component, with statistically significant and clinically relevant improvements over placebo for the primary endpoints, across all doses tested. Cebranopadol displayed additional beneficial effects including improved sleep and functionality.	[35]
Patients with moderate-to-severe cancer-related pain	≥18 years of age Male/ Female	Oral	200–1000 μg (*n* = 65) once daily treatment period: 44 days	Average amount of daily rescue medication intake.	Cebranopadol was effective in the dose range tested (200–1000 mcg) in a patient population with chronic pain related to cancer. Most used doses of cebranopadol were ≤800 mcg Cebranopadol was noninferior and superior to morphine PR on reduction of daily rescue medication intake over the last 2 weeks of treatment (the primary endpoint).	[36]

**Table 3 molecules-27-03987-t003:** Treatment scheme for the naloxone-precipitated withdrawal jumping model [31].

Day	1	2	3	4	5
Administration	1	2	1	2	1
Morphine mg/kg, ip	1	2	4	8	16
Cebranopadol mg/kg, ip	0.04	0.08	0.16	0.32	0.64

## Data Availability

Not applicable.

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
