# Peer review of "Cebranopadol as a Novel Promising Agent for the Treatment of Pain"

_molecules, 2022, doi:10.3390/molecules27133987_

Round 1

Reviewer 1 Report

 Cebranopadol is a novel, first-in-class analgesic that acts as a NOP receptor and mixed opioid receptor agonist. The compound was has beeb studied in approximately 2,000 patients worldwide having completed Phase II clinical trials as well as a Phase III study.
 Authors timely reviewed the compounds by focusing on its sites of action, its pharmacokinetics and also clinical trials to further explore the role as an novel pain killer. The review is worth showing and reading readers in the Molecules especially researchers for conducting the research of development of novel analgesics including opioids and non-opioids.

Although there are some typos and grammatical errors in the ms, it can be revised.

Author Response

We thank the Reviewer 1 and Reviewer 2 for the thoughtful and critical review of this manuscript

The article has been modified to reflect your concerns, including linguistic proofreading. All changes made in the manuscript are in bold.

Reviewer 1.

 “Cebranopadol is a novel, first-in-class analgesic that acts as a NOP receptor and mixed opioid receptor agonist. The compound was has beeb studied in approximately 2,000 patients worldwide having completed Phase II clinical trials as well as a Phase III study.
 Authors timely reviewed the compounds by focusing on its sites of action, its pharmacokinetics and also clinical trials to further explore the role as an novel pain killer. The review is worth showing and reading readers in the Molecules especially researchers for conducting the research of development of novel analgesics including opioids and non-opioids.

Although there are some typos and grammatical errors in the ms, it can be revised.”

Answer: Thank you for your comment. The manuscript was modified including linguistic proofreading.

Reviewer 2.

“The present manuscript reviews the current literature regarding cebranopadol, a novel analgesic that activates both opioid receptors and NOP receptors. The authors have put together a nice review of the literature; however, there is evidence showing that similar to opioids cebranopadol may be reinforcing, which is missing in this review. In addition, the manuscript needs to be reviewed critically as it contains typos and grammatical errors. I have highlighted some of those in the attached PDF document. My specific comments are listed below”

  1. Please provide a reference for the statement on lines 57-59 and 172-174.

Answer: Thank you for pointing this out. The reference was added.

  1. While the drug is long-acting and it takes about 7 hours for its clearance, there is limited information on whether the drug would accumulate in the body after repeated dosing or in patients with liver and kidney issues.

Answer: Thank you for bringing to our notice how relevant this task is. After delving into the topic, we searched for the information in the manuscripts we used. We found additional information about accumulation of the drug, as well as its safety. Unfortunately, we did not find any relevant information about the use of the drug in treating patients with liver and kidney issues. We added extra information which mainly responds to this question

  1. Table 3 needs to be more comprehensive. It should include more information than provided, such as which species of animals were used, whether it was male or female, the dose of cebranopadol, and the outcome of the study. The table should have different cells for this information and should be presented in the landscape format.

Answer: Thank you for your comment. The table was change according to your suggestions.

  1. Cebranopadol has been reported to reduce cocaine intake but there is not much detail about this. How the drug reduces cocaine intake, whether its NOP action is responsible for this or MOP?

Answer: Thank you for this suggestion. We added information which specifies this action of cebranopadol.

Reviewer 2 Report

The present manuscript reviews the current literature regarding cebranopadol, a novel analgesic that activates both opioid receptors and NOP receptors. The authors have put together a nice review of the literature; however, there is evidence showing that similar to opioids cebranopadol may be reinforcing, which is missing in this review. In addition, the manuscript needs to be reviewed critically as it contains typos and grammatical errors. I have highlighted some of those in the attached PDF document. My specific comments are listed below:

Major

  1. Please provide a reference for the statement on lines 57-59 and 172-174.
  2. While the drug is long-acting and it takes about 7 hours for its clearance, there is limited information on whether the drug would accumulate in the body after repeated dosing or in patients with liver and kidney issues.
  3. Table 3 needs to be more comprehensive. It should include more information than provided, such as which species of animals were used, whether it was male or female, the dose of cebranopadol, and the outcome of the study. The table should have different cells for this information and should be presented in the landscape format.
  4. Cebranopadol has been reported to reduce cocaine intake but there is not much detail about this. How the drug reduces cocaine intake; whether its NOP action is responsible for this or MOP?
  5. The manuscript should be critically reviewed for typos and grammatical errors. For specific comments, please see the highlighted sections of the attached PDF.
  6.  

Author Response

(The authors gave the same response as above.)

Round 2

Reviewer 2 Report

The authors addressed the concerns of the previous version. However, they need to critically review the final version to ensure there is no typos or grammatical errors.

Author Response

Dear Editor, 

Thank You for Your comments. The manuscript has been revised according to Your indications. 

Best regards, 

Wojciech M. Ziemichód